# Epithelial cell-turnover ensures robust coordination of tissue growth in *Drosophila* ribosomal protein mutants

**Nanami Akai[1,2]⊗, Shizue Ohsawa[1,2]⊗\*, Yukari Sando[1], Tatsushi Igaki[1]\***

**1** Laboratory of Genetics, Graduate School of Biostudies, Kyoto University, Kyoto, Japan, **2** Group of Genetics, Division of Biological Science, Graduate School of Science, Nagoya University, Nagoya, Japan

⊗ These authors contributed equally to this work.
\* ohsawa.shizue@d.mbox.nagoya-u.ac.jp (SO); igaki@lif.kyoto-u.ac.jp (TI)

**Data Availability Statement:** All relevant data are within the manuscript and its Supporting Information files.

## Abstract

Highly reproducible tissue development is achieved by robust, time-dependent coordination of cell proliferation and cell death. To study the mechanisms underlying robust tissue growth, we analyzed the developmental process of wing imaginal discs in *Drosophila Minute* mutants, a series of heterozygous mutants for a ribosomal protein gene. *Minute* animals show significant developmental delay during the larval period but develop into essentially normal flies, suggesting there exists a mechanism ensuring robust tissue growth during abnormally prolonged developmental time. Surprisingly, we found that both cell death and compensatory cell proliferation were dramatically increased in developing wing pouches of *Minute* animals. Blocking the cell-turnover by inhibiting cell death resulted in morphological defects, indicating the essential role of cell-turnover in *Minute* wing morphogenesis. Our analyses showed that *Minute* wing discs elevate Wg expression and JNK-mediated Dilp8 expression that causes developmental delay, both of which are necessary for the induction of cell-turnover. Furthermore, forced increase in Wg expression together with developmental delay caused by ecdysone depletion induced cell-turnover in the wing pouches of non-*Minute* animals. Our findings suggest a novel paradigm for robust coordination of tissue growth by cell-turnover, which is induced when developmental time axis is distorted.

## Author summary

Animal development can be disturbed by various stimuli such as genetic mutations, environmental fluctuations, and physical injuries. However, animals often accomplish normal tissue growth and morphogenesis even in the presence of developmental perturbations. *Drosophila Minute* mutants, a series of fly mutants for a ribosomal protein gene, show significantly prolonged larval period but develop into essentially normal flies. We found an unexpected massive cell death and subsequent compensatory cell proliferation in developing wing discs of *Minute* animals. This 'cell-turnover' was essential for normal wing morphogenesis in *Minute* flies. We found that the cell-turnover was induced by elevated Wg expression in the wing pouch and JNK-mediated Dilp8 expression that causes

**Funding:** This work was supported by Grant-in-Aid for Scientific Research (A) (Grant No. 16H02505) to T.I, Grant-in-Aid for Scientific Research on Innovative Areas (Grant Nos. 15H05862, 26114002, 20H04866) to S.O and T.I, Grant-in-Aid for challenging Exploratory Research (Grant No. 19K22423) to S.O, the Naito Foundation to S.O and T.I, the Takeda Science Foundation to S.O and T.I, Japan Agency for Medical Research and Development (Project for Elucidating and Controlling Mechanisms of Aging and Longevity, Grant Number 17938731) to T.I., Inamori Foundation to S.O., Toray Science Foundation to S. O., Senri Life Science Foundation to S.O., Yamada Science Foundation to S.O. and the Mitsubishi Foundation to S.O. The funders had no role in study design, data collection and analysis, decision to publish, or preparation of the manuscript.

**Competing interests:** The authors have declared that no competing interests exist.

developmental delay. Indeed, cell-turnover was reproduced in non-*Minute* animals' wing discs by overexpressing Wg using the *wg* promoter together with developmental delay caused by ecdysone depletion. Our findings propose a novel paradigm for morphogenetic robustness by cell-turnover, which ensures normal wing growth during the abnormally prolonged larval period, possibly by creating a flexible cell death and proliferation platform to adjust cell numbers in the prospective wing blade.

## Introduction

Multicellular organisms accomplish normal tissue growth in a highly reproducible manner even in the presence of various perturbations such as genetic mutations, environmental fluctuations, and physical injuries [1]. To achieve such robust development, cells in the developing tissue must behave in a plastic and coordinated manner to correct perturbations and ensure time-dependent morphogenesis. For instance, when abnormally massive cell death occurs in developing *Drosophila* imaginal epithelia, dying cells trigger compensatory proliferation of surrounding viable cells by secreting growth factors to ensure normal morphogenesis [2–4]. In addition, in response to a physical injury induced in *Drosophila* imaginal epithelia, the larval period is extended to allow injured tissue time to regenerate before entering metamorphosis [5]. During the extended larval period, other intact tissues stop growing until the injured or delayed tissue catches up in development [6]. Thus, multicellular organisms possess buffering mechanisms against perturbations in the developmental time axis through systemic and/or short-range cell-cell communications that govern cell proliferation and cell death.

Reproducible animal morphogenesis is ensured by the precise progression of developmental timing, and thus a distortion in the developmental time axis potentially causes developmental defects. It has been well documented that *Drosophila* mutants heterozygous for ribosomal proteins, called *Minute*/+ (*M*/+) mutants, show significant delay in the progression of their larval period (~1.5 times longer than wild-type larvae) [7]. Intriguingly, despite the significant delay in their development, *M*/+ animals are essentially normal flies without any significant morphological defects except for relatively thinner bristles than wild-type [8]. This suggests that *M*/+ animals must enforce morphogenetic robustness during the abnormally prolonged larval period. However, the mechanism by which *M*/+ animals exert morphogenetic robustness against the developmental time distortion remained unknown. Here, we find evidence that both cell death and compensatory cell proliferation are dramatically increased in the *M*/+ wing imaginal epithelium, which are essential for robust wing development in *M*/+ animals. Our data suggest that massive cell-turnover creates a flexible cell death/proliferation platform to adjust cell numbers in the developing tissue.

## Results

### Massive cell-turnover occurs during robust morphogenesis of *M*/+ tissue

To explore how animals ensure robust coordination of tissue growth, we analyzed cellular behaviors in the developing *M*/+ wing imaginal disc. It has been well documented by clonal analysis that the growth rate of *M*/+ cell clones is lower than that of wild-type clones in developing wing discs [9]. The lower growth rate of *M*/+ animals has been largely assumed to stem from a lower cell division rate of *M*/+ cells, without any defects in cell proliferation and cell death. Surprisingly, however, we found that cell mitoses were significantly increased in the wing pouch of *RpS3*/+, one of the *M*/+ mutants, compared to wild-type, as visualized by M

phase marker phospho-Histone H3 (Fig 1A and 1B, quantified in Fig 1C; the wing pouch is pale green-marked oval domain that becomes the adult wing blade) and EdU labeling for S phase cells (Fig 1D and 1E, quantified in Fig 1F) (hereafter, developmental stages of *M/+* and wild-type animals examined were adjusted to the wandering larval stage by using the *sgsΔ3-GFP* reporter [10]). Similar increased cell proliferation was also observed in the wing pouches of other *M/+* strains such as *RpS17/+* and *RpL19/+* (S1A and S1B Fig, quantified in S1C Fig). We further examined this using the cell cycle monitoring probe S/G2/M-Green that labels cells in S/G2/M phases [11,12], which showed that the number of cells in either S/G2 or M phase significantly increased in the *RpS3/+* pouch compared to wild-type (Fig 1G and 1H). Furthermore, time-lapse imaging of cultured wing discs confirmed that the number of dividing cells was significantly increased in the *RpS3/+* pouch (S2 Movie, quantified in Fig 1I), while the interval between S/G2 and M phases in the dividing cells in the *RpS3/+* pouch was comparable to wild-type (S1 Movie, quantified in Fig 1J). These data reveal that cells in slow-growing *M/+* wing pouch proliferate more rapidly than cells in wild-type wing pouch.

The fact that *M/+* cells have a higher cell division rate suggests that there must be a compensatory mechanism that adjusts cell number in developing *M/+* wing pouch. Strikingly, we found that massive cell death occurred in the developing *RpS3/+* wing pouch, while only a few dying cells were detected in the wild-type pouch (Fig 1K and 1L, arrowheads, quantified in Fig 1M; Fig 1N and 1O, quantified in Fig 1P), using the antibody against cleaved-Dcp1 or CD8-PARP-Venus probe that visualizes caspase activity in the imaginal disc [13–20]. Similar massive cell death was also observed in the wing pouches of other *M/+* strains, such as *RpS17/+* and *RpL19/+* animals (S1D and S1E Fig). The incidence of cell death in the *M/+* wing pouch increased as development proceeded and peaked during the middle to late 3rd instar (S1F and S1G Fig). Consistent with these observations, blocking effector caspases by overexpression of baculoviral p35 [21] in the wing pouch resulted in wing disc overgrowth (S2C Fig, compared to S2A and S2B Fig) and severe morphological defects with blisters in the adult *RpS3/+* wing (Fig 2C, compared to Fig 2A and 2B, quantified in Fig 2F), while p35-overexpression alone had no effect on wild-type or RpS3-rescued *RpS3/+* wing development (Fig 2D and 2E, quantified in Fig 2F). This morphological defect was likely due to the generation of "undead cells", which produce secreted growth factors such as Wingless (Wg; a Wnt homolog) and Decapentaplegic (Dpp, a Dpp homolog) via elevated initiator caspase activity [21–23]. p35-overexpression in the wing pouch similarly resulted in severe morphological defects with blisters in other *M/+* such as *RpS17/+* and *RpL19/+* (S2D and S2E Fig, quantified in S2F Fig). This further demonstrates massive cell death occurring in *M/+* wing pouch. Time-lapse imaging of *ex vivo* wing morphogenesis indicated that the *RpS3/+* wing pouch expressing p35 overgrew and therefore failed to appose the ventral and dorsal surfaces of the wing blade correctly (S2G and S2H Fig, quantified in S2I Fig; S3–S5 Movies), which is consistent with the adult wing blister phenotype [24].

The increased cell death and proliferation suggests that *M/+* cells in the wing pouch trigger apoptosis-induced compensatory cell proliferation [3,4]. Indeed, we found that the increased cell proliferation depended on the induction of cell death, as cell proliferation was significantly reduced by blocking cell death in the *RpS3/+* wing pouch using heterozygous chromosomal deletion *H99* (which removes cell death genes *reaper*, *hid*, and *grim*) [25–27] (Fig 2G, quantified in Fig 2I) or dominant-negative form of the *Drosophila* initiator caspase Dronc (Dronc[DN]) [28] (Fig 2H, quantified in Fig 2I), as assessed by phospho-Histone H3 staining (Fig 2J and 2K, quantified in Fig 2L). These data indicate that the *M/+* wing pouch exhibits greatly enhanced cell-turnover that is induced by massive apoptosis and subsequent compensatory cell proliferation.

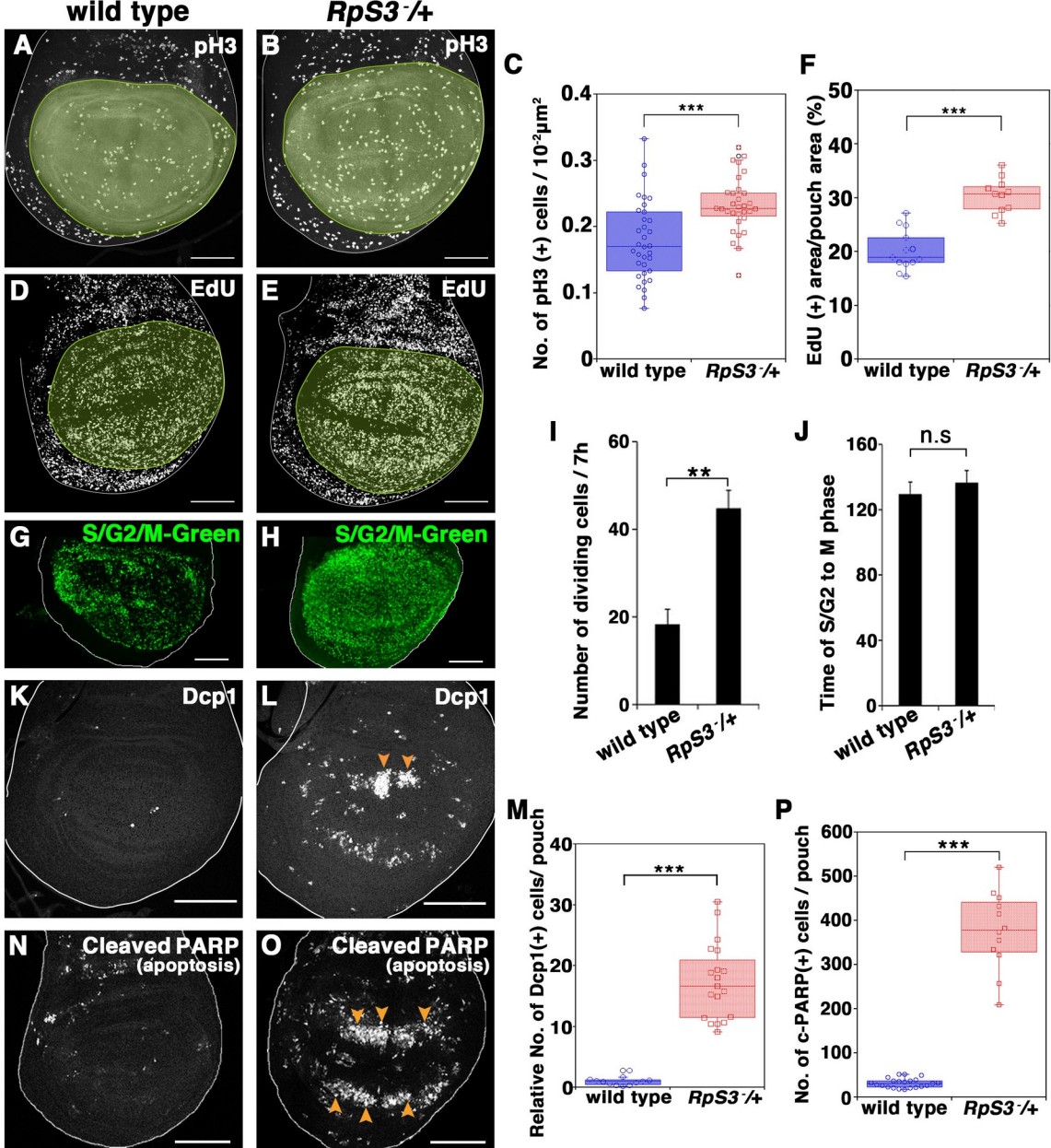

**Fig 1. Massive cell-turnover occurs during wing development in *M/+* animals. (A and B)** Wing disc of wild-type (A) or *RpS3/+* (B) flies were stained with anti-phospho-histone H3 (pH3) (Ser10) antibody (white). Wing pouches were marked in pale green. Scale bar, 100 μm. **(C)** Boxplot with dots representing pH3 positive cells in the pouch in genotypes shown in (A) (n = 34, number of wing pouches) and (B) (n = 31). Error bars, SEM; ***, p<0.001; non-parametric Mann-Whitney *U*-test. **(D and E)** S-phase cells in the wing discs of wild-type (D) or *RpS3/+* (E) flies were visualized by EdU staining (nuclei of proliferating cells) (white). Wing pouches were marked in pale green. Scale bar, 100 μm. **(F)** Boxplot with dots representing the EdU-positive area per wing pouch in genotypes shown in (D) (n = 13, number of wing pouches) and (E) (n = 11). Error bars, SEM; ***, p<0.001; non-parametric Mann-Whitney *U*-test. **(G and H)** Dividing cells were visualized by the S/G2/M-Green probe. Wing disc of *nub-Gal4*, UAS-S/G2/M-Green (G) or *RpS3/+, nub-Gal4*, UAS-S/G2/M-Green (H) is shown. Scale bar, 100 μm. **(I)** Quantification of the manually counted number of dividing cells in the wing pouch of genotypes shown in (G) (n = 7, number of wing pouches) and (H) (n = 10). Error bars, SEM; **, p<0.01; non-parametric Mann-Whiteney *U*-test. **(J)** Duration time (min) of S/G2 to M phase in cells in the wing pouch of wild-type (n = 19, number of wing pouches) or *RpS3/+* (n = 42) larvae were manually measured. Time-laps imaging was performed for 7 hours. Error bars, SEM; n.s, not significant; non-parametric Mann-Whiteney *U*-test. **(K and L)** Dying cells in the wing disc were visualized by anti-Dcp1 antibody staining. Wing disc of wild-type (K) or *RpS3/+* (L) is shown. Arrowheads indicate massive cell death occurred along the D/V axis in the wing pouch. Scale bar, 100 μm. **(M)** Boxplot with dots representing cleaved-Dcp-1-positive dying cells per pouch in genotypes shown in (K) (n = 12, number of wing pouches), and (L) (n = 19). Error bars, SEM; ***, p<0.001; non-parametric Mann-

Whitney *U*-test. Scale bar, 100 μm. **(N and O)** The activated-caspase-3 indicator CD8-PARP-Venus was expressed in wild-type (N) or *RpS3*/+ (O) flies, and dying cells in the wing disc were visualized by anti-cleaved PARP staining (white). Arrowheads indicate massive cell death. The nuclei were visualized by DAPI staining (blue). Scale bar, 100 μm. **(P)** Boxplot with dots representing cleaved-PARP-positive dying cells per pouch in genotypes shown in (A) (n = 22, number of wing pouches) and (B) (n = 12). Error bars, SEM; ***, p<0.001; non-parametric Mann-Whitney *U*-test. Genotypes are as follows: *nub-Gal4/+; UAS-CD8-PARP-Venus/+* (A, D), *nub-Gal4, UAS-CD8-PARP-Venus, RpS3^{Plac92}/+* (B, E), *nub-Gal4/+; UAS-S/G2/M-Green/+* (G), *nub-Gal4/+; UAS-S/G2/M-Green/RpS3^{Plac92}* (H), *w^{1118}* (K), *FRT82B, Ubi-GFP, RpS3^{Plac92}/+* (L), *Actin-Gal4, UAS-CD8-PARP-Venus/+* (N), and *Actin-Gal4, UAS-CD8-PARP-Venus/ FRT82B, Ubi-GFP, RpS3^{Plac92}* (O).

## Cell-turnover is required for normal wing morphogenesis in *M*/+ animals

We next investigated whether this cell-turnover is essential for normal wing morphogenesis in *M*/+ animals. Significantly, reducing cell death and compensatory cell proliferation, namely cell-turnover, in the *RpS3*/+ wing pouch by introducing *H99*/+ or by overexpressing Dronc^{DN} resulted in morphological defects in the adult wing, while *H99*/+ or Dronc^{DN} alone had no effect on wing development (Fig 2M and 2N). These data suggest that *M*/+ wing disc induces cell-turnover to ensure robust wing morphogenesis.

## Aberrant Wg signaling gradient is essential for massive cell-turnover

We next sought to address how the *M*/+ wing pouch induces massive cell-turnover. We noticed that most cell death occurred in the *RpS3*/+ wing pouch along the DV axis at a constant distance from the DV boundary, where the morphogen Wingless (Wg; a Wnt homolog) is expressed (Fig 3A and 3B). Wg signaling is required in the wing pouch for cell specification, tissue growth, patterning, and morphogenesis during wing development (Reviewed in [29,30]). The expression pattern of Wg was indistinguishable between *RpS3*/+ and wild-type wing pouches (Fig 3A"and 3B"). Notably, we found that the Wg expression level was elevated in the *M*/+ wing pouch, as genetic rescue of RpS3 in the posterior compartment of the *RpS3*/ + wing disc resulted in decreased Wg transcription and protein expression compared to the anterior *RpS3*/+ control (S3A and S3D Fig, quantified in S3C and S3F Fig). In addition, Wg signaling activity was elevated much more broadly in the *RpS3*/+ pouch compared to the same-stage wild-type (wondering larval stage), as assessed by the *nmo-lacZ* reporter [31] (Fig 3C and 3D). The elevation of Wg signaling activity in the *RpS3*/+ wing pouch was further confirmed by the genetic rescue experiment whereby RpS3 was overexpressed in the posterior compartment of the *RpS3*/+ disc, which showed decreased expression of Wg activity reporters *nmo-lacZ* [32] and *nkd-lacZ* [33] compared to the anterior control (Fig 3F and 3H), while overexpression of RpS3 alone did not affect Wg signaling activity in the wild-type wing disc (Fig 3E and 3G). Interestingly, the areas of massive cell death in the *RpS3*/+ wing pouch always corresponded to the areas of relatively lower Wg signaling activity (Fig 3D). This raises the possibility that massive cell death could be induced by Wg-dependent cell competition, a phenomenon by which cells with higher Wg signaling activity eliminate neighboring cells with lower Wg signaling activity in the wing disc [34]. We assumed that in the *RpS3*/+ wing pouch, ectopically upregulated Wg signaling at a distant site from the DV boundary might induce cell death in neighboring distal cells with relatively lower Wg signaling activity. Indeed, cell death was markedly suppressed when the aberrant Wg signaling gradient was reduced either by deleting one copy of the *wg* gene or by overexpressing Wg in the entire wing pouch (Fig 3I and 3J, quantified in Fig 3L). Furthermore, reduction of the Wg signaling gradient by downregulating *dishevelled* (*dsh*) or *dtcf*, downstream effectors of Wg signaling [35–39], in the *RpS3*/ + wing pouch also strongly suppressed cell death (Fig 3K, quantified in Fig 3L; S3G and S3H Fig, quantified in Fig S3K). Similarly, gentle upregulation of Wg signaling activity in the entire wing pouch by introducing one copy of *wg^{Gla-1}*, a gain of function allele of *wg*, or by mildly

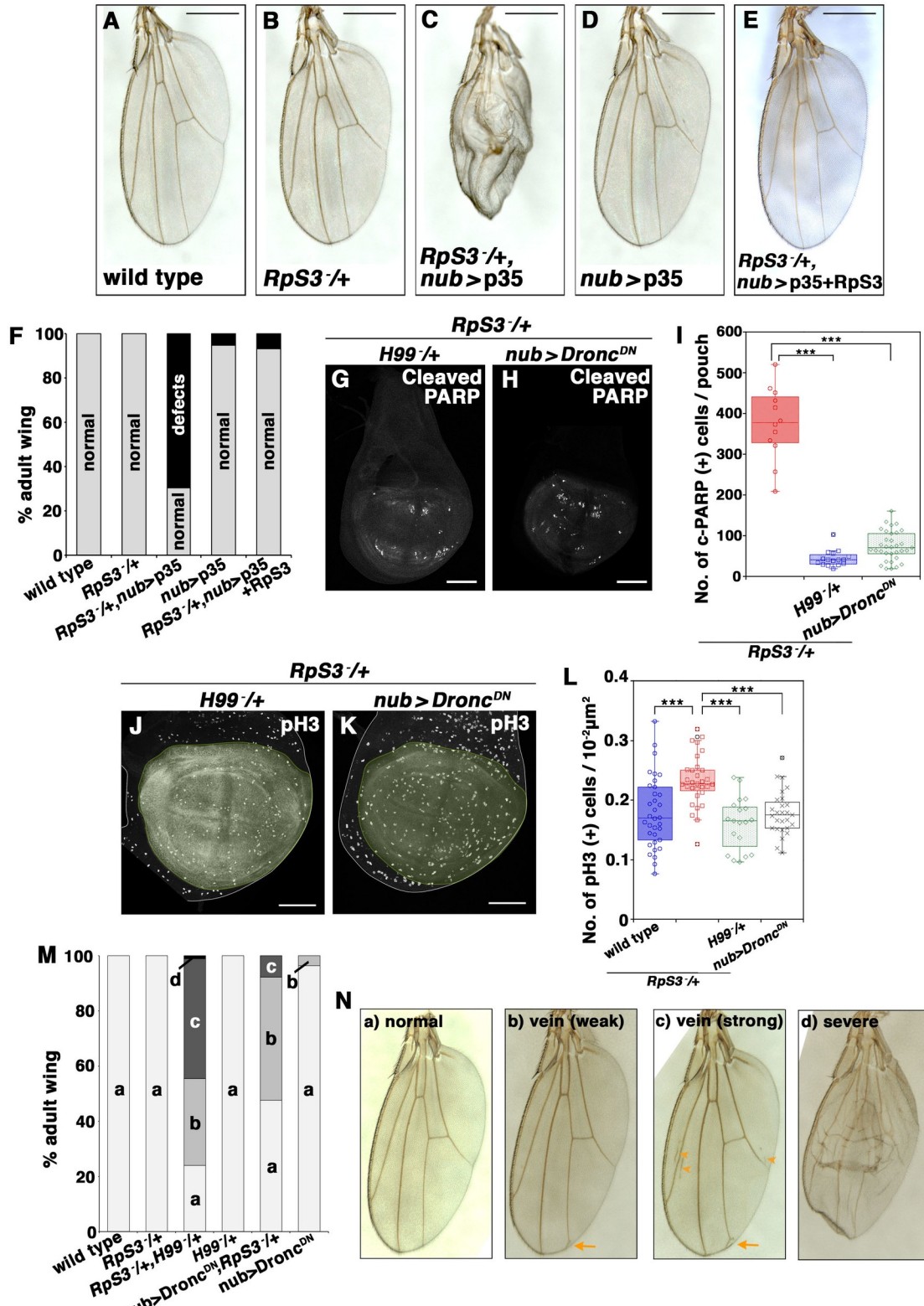

**Fig 2. Epithelial cell-turnover is essential for robust wing development in *M*/+ animals. (A-E)** Adult wings of wild-type (A), *RpS3/+* (B), *RpS3/+, nub-Gal4*, UAS-p35 (C), *nub-Gal4*, UAS-p35 (D), or *RpS3/+, nub-Gal4*, UAS-p35, UAS-RpS3 (E) fly. Scale bar, 500 μm. **(F)** The rate of defective wings in the genotypes shown in (A) (n = 150), (B) (n = 271), (C) (n = 128), (D) (n = 125), and (E) (n = 44). **(G and H)** Dying cells were visualized by anti-cleaved PARP staining (white) in the wing discs of *RpS3/+, H99/+*

(G) or *RpS3/+, nub-Gal4, UAS-Dronc$^{DN}$* (H). Undead cells were not generated in these situations. Scale bar, 100 μm. **(I)** Boxplot with dots representing cleaved-PARP-positive dying cells per pouch in genotypes shown in (Fig 1B) (n = 12, number of wing pouches), (G) (n = 16), and (H) (n = 33). Error bars, SEM; ***, p<0.001; non-parametric Mann-Whitney *U*-test. **(J and K)** Wing disc of *RpS3/+, H99/+* (J) or *RpS3/+, nub-Gal4, UAS-Dronc$^{DN}$* (K) were stained with anti-phospho-histone H3 (pH3) (Ser10) antibody (white). Wing pouches are marked in pale green. Scale bar, 100 μm. **(L)** Boxplot with dots representing pH3 positive cells per pouch in genotypes shown in (Fig 1A) (n = 34, number of wing pouches), (Fig 1B) (n = 31), (J) (n = 19), and (K) (n = 27). Error bars, SEM; ***, p<0.001; non-parametric Mann-Whitney *U*-test. **(M)** The rate of defective wings in the genotypes of wild-type (n = 70), *RpS3/+* (n = 86), *RpS3/+, H99/+* (n = 83), *H99/+* (n = 100), *RpS3/+, nub-Gal4, UAS-Dronc$^{DN}$* (n = 63), and *nub-Gal4, UAS-Dronc$^{DN}$* (n = 109). **(N)** Adult wing phenotypes were classified as following five types: (a) normal, (b) vein (weak) (weakly having a brunched vein in the point of "landmark 14" (arrow) [69]), (c) vein (strong) (having other additional vein phenotypes (arrowheads) in addition to "landmark 14"), and (d) severe. Genotypes are as follows: *w$^{1118}$* (A), *FRT82B, Ubi-GFP, RpS3$^{Plac92}$/+* (B), *nub-Gal4/+; FRT82B, Ubi-GFP, RpS3$^{Plac92}$/UAS-p35* (C), *nub-Gal4/+; UAS-p35/+* (D), *nub-Gal4/UAS-RpS3; UAS-p35/ FRT82B, Ubi-GFP, RpS3$^{Plac92}$* (E), *nub-Gal4/+; UAS-CD8-PARP-Venus, RpS3$^{Plac92}$/Df(3L)H99* (G, J), *nub-Gal4/+; UAS-CD8-PARP-Venus, RpS3$^{Plac92}$/UAS-Dronc$^{DN}$* (H, K), *nub-Gal4/+; UAS-CD8-PARP-Venus/+* (wild-type), *nub-Gal4/+; UAS-CD8-PARP-Venus, RpS3$^{Plac92}$/+* (*RpS3$^-$/+*), *nub-Gal4/+; UAS-CD8-PARP-Venus, RpS3$^{Plac92}$/Df(3L)H99* (*RpS3$^-$/+, H99/+*), *nub-Gal4/+; UAS-CD8-PARP-Venus/Df(3L)H99* (*H99/+*), *nub-Gal4/+; UAS-CD8-PARP-Venus, RpS3$^{Plac92}$/UAS-Dronc$^{DN}$* (*RpS3$^-$/ +, nub>Dronc$^{DN}$*), and *nub-Gal4/+; UAS-CD8-PARP-Venus/UAS-Dronc$^{DN}$* (*nub>Dronc$^{DN}$*) in (M).

overexpressing an active form of Armadillo$^{S10}$ (Arm$^{S10}$) [40] significantly suppressed cell death (S3I and S3J Fig, quantified in S3K Fig). These data suggest that an aberrant Wg signaling gradient generated in the *M/+* wing pouch is required for the induction of massive cell death.

## JNK-Dilp8-mediated developmental delay is required for massive cell-turnover

From the results presented so far, we postulated that aberrant Wg gradient through upregulation of endogenous Wg expression could be sufficient to cause massive cell death in the wing pouch. However, Wg-overexpression using the endogenous *wg* promoter did not induce cell death in the wing pouch (S4A and S4B Fig, quantified in S4C Fig). This indicates that the *M/+* wing pouch must induce an additional event(s) essential for causing massive cell death. A strong candidate for the missing event is developmental delay, as the prolonged larval period of *M/+* animals (about 40 hours longer than that of wild-type [7–9]) may enhance the formation of an aberrant Wg signaling gradient. It has been reported that *M/+* imaginal discs upregulate *Drosophila insulin-like peptide 8* (*dilp8*), which causes a developmental delay [41,42]. Indeed, in *RpS3/+* animals, *dilp8* was upregulated in various imaginal discs, especially in the wing pouch (Fig 4A and 4B). In addition, consistent with previous reports [42–45], blocking JNK signaling by overexpression of the JNK phosphatase Puckered (Puc) abolished *dilp8* upregulation in the *RpS3/+* wing pouch (S4D Fig). Consistent with these results, developmental delay in the larval period of *RpS3/+* animals was partially suppressed by reducing *dilp8* gene dosage or downregulating *dilp8* expression in the wing pouch, as well as by blocking JNK signaling in the wing pouch (Fig 4C). Moreover, the *RpS3/+* wing pouch indeed elevated JNK activity, as genetic rescue of RpS3 expression in the posterior compartment of the *RpS3/+* disc resulted in decreased expression of JNK sensor TRE-DsRed [46] and JNK target genes *misshapen* (*msn*) [47] compared to the anterior control (S4F and S4H Fig); overexpression of RpS3 on its own did not affect JNK activity in wild-type wing disc (S4E and S4G Fig). These data indicate that elevated JNK activity in the *M/+* wing pouch causes Dilp8-mediated developmental delay in the *M/+* larvae. The developmental delay in *M/+* animals, however, could be initially triggered by additional unknown causes, since developmental delay was not fully rescued by blocking the JNK-Dilp8 axis in *M/+* wing discs.

We thus examined whether Dilp8-dependent developmental delay contributes to the induction of massive cell death in the *M/+* wing pouch. It has been reported that Dilp8 causes a developmental delay by inhibiting ecdysone biosynthesis in the prothoracic gland [41,42].

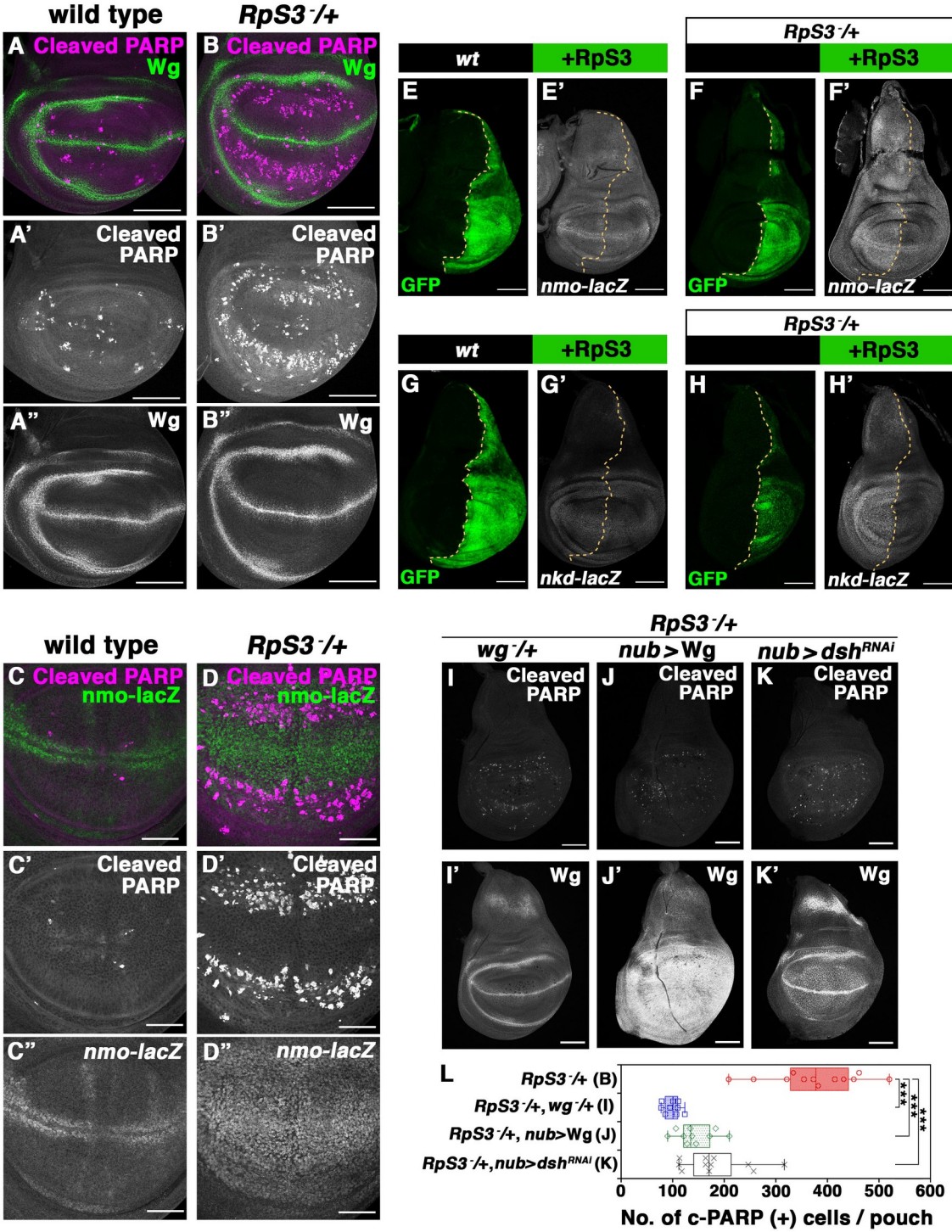

**Fig 3. Aberrant Wg signaling gradient is essential for the induction of massive cell-turnover. (A-B")** The relative position between dying cells (cleaved-PARP, magenta) and Wg expression (anti-Wg, green) in the wild-type (A) or *RpS3/+* (B) wing pouch. Scale bar, 100 μm. **(C-D")** The relative position between dying cells (cleaved-PARP, magenta) and Wg signaling activity (anti-β-galactosidase, green) in the *nmo-LacZ/+* (C) or *RpS3/+, nmo-LacZ/+* (D) wing pouch. These images are Z-stacked images from 8 (C) and 8 (D) confocal images (3.02 μm and 2.39 μm, respectively) for whole wing pouch that is acquired every 1.01 μm or 0.8 μm, respectively. Scale bar, 50 μm. **(E-F')** RpS3 was overexpressed in the posterior compartment of wing discs of *nmo-lacZ/+* (E) or *RpS3/+, nmo-lacZ/+* (F) flies using the *en-Gal4* driver. *nmo* expression was visualized by anti-β-galactosidase staining (white). Scale bar, 100 μm. **(G-H')** RpS3 was overexpressed in the posterior compartment of wing discs of *nkd-lacZ/+* (G) or *RpS3/+, nkd-lacZ/+* (H) flies using the *en-Gal4* driver. *nkd* expression was visualized by anti-β-galactosidase staining (white). Scale bar, 100 μm. **(I-K')** Dying

cells were visualized by anti-cleaved PARP staining (white) in the wing discs of *RpS3/+*, *wg/+* (I), *RpS3/+*, *nub-Gal4*, UAS-Wg (J), or *RpS3/+*, *nub-Gal4*, *UAS-dsh-RNAi* (K) flies expressing CD8-PARP-Venus. Wg expression was visualized by anti-Wg staining (white). Scale bar, 100 μm. **(L)** Boxplot with dots representing cleaved-PARP-positive dying cells per pouch in genotypes shown in (B) (n = 12, number of wing pouches), (I) (n = 9), (J) (n = 10), and (K) (n = 11). Error bars, SEM; \*\*\*, p<0.001; non-parametric Mann-Whitney *U*-test. Genotypes are as follows: *nub-Gal4/+; UAS-CD8-PARP-Venus/+* (A), *nub-Gal4, UAS-CD8-PARP-Venus, RpS3$^{Plac92}$/+* (B), *nub-Gal4/+; UAS-CD8-PARP-Venus/nmo$^{P1}$* (C), *nub-Gal4/+; UAS-CD8-PARP-Venus, RpS3$^{Plac92}$/nmo$^{P1}$* (D), *en-Gal4, UAS-GFP/+; UAS-RpS3/nmo$^{P1}$* (E), *en-Gal4, UAS-GFP/+; RpS3$^{Plac92}$, UAS-RpS3/nmo$^{P1}$* (F), *en-Gal4, UAS-GFP/+; UAS-RpS3/nkd$^{04869a}$* (G), *en-Gal4, UAS-GFP/+; RpS3$^{Plac92}$, UAS-RpS3/nkd$^{04869a}$* (H), *nub-Gal4/wg$^{l-8}$; UAS-CD8-PARP-Venus, RpS3$^{Plac92}$/+* (I), *nub-Gal4/+; UAS-CD8-PARP-Venus, RpS3$^{Plac92}$/UAS-Wg* (J), and *nub-Gal4/UAS-dsh-RNAi; UAS-CD8-PARP-Venus, RpS3$^{Plac92}$/+* (K).

Indeed, the peak of the active form of ecdysone 20-Hydroxyecdysone (20E) level was delayed in the *RpS3/+* 3rd instar larvae, compared to wild-type (Fig 4D). In addition, massive cell death as well as the developmental delay of *RpS3/+* larvae [48] were rescued by feeding larvae with 20E (Fig 4E and 4F, quantified in Fig 4G). In addition, reduction of *dilp8* expression by heterozygosity for the *dilp8* gene or by *dilp8-RNAi* strongly suppressed cell death (Fig 4H and 4I, quantified in Fig 4K). Similar results were obtained when *dilp8* expression was blocked by overexpression of Puc (Fig 4J, quantified in Fig 4K). These data suggest that the developmental delay caused by the JNK-Dilp8 module contributes to the induction of cell-turnover in the *M/ +* wing pouch.

## Developmental delay in conjunction with Wg upregulation causes massive cell-turnover

Finally, we asked whether massive cell-turnover could be reproduced in a non-*M/+* wing pouch by inducing both developmental delay and Wg upregulation. The temperature-sensitive *ecdysoneless* (*ecd*) mutant flies fail to produce ecdysone and therefore have a prolonged larval period upon heat-shock treatment [49]. We found that the number of dying cells was moderately increased in *ecd* mutant wing discs (Fig 4M, compare to Fig 4L, quantified in Fig 4O). Strikingly, a forced increase in endogenous Wg expression in conjunction with developmental delay caused massive cell death in the wing pouch along the DV axis (Fig 4N, quantified in Fig 4O), which was accompanied by increased cell mitoses (S4I and S4J Fig, quantified in S4K Fig). In addition, depletion of ecdysone in wild-type 3rd instar larvae using the fly food with *erg-2* mutant yeast (*erg-2Δ*) [50,51] resulted in prolonged larval period and significantly increased number of dying cells in the wing disc (S4L Fig, compared to S4A Fig, quantified in S4N Fig). A forced increase in endogenous Wg expression in conjunction with *erg-2Δ*-induced developmental delay resulted in more increased cell death in the wing pouch along the DV axis (S4M Fig, quantified in S4N Fig). Furthermore, induction of homozygous mutations in an apicobasal polarity gene *scribble* (*scrib$^{-/-}$*) in the entire eye discs, which resulted in the formation of tumors and extended larval period [52], significantly increased the number of dying cells in the wing discs (S4O Fig, compared to S4A Fig, quantified in S4Q Fig). A forced increase in endogenous Wg expression in conjunction with the tumor-induced developmental delay resulted in more increased cell death in the wing pouch (S4P Fig, quantified in S4Q Fig). Together, these data suggest that *M/+* wing imaginal epithelium undergoes massive cell-turnover through the combination of upregulated Wg expression and developmental delay, thereby ensuring normal wing morphogenesis (Fig 5).

## Discussion

Our genetic study of *Drosophila M/+* mutants proposes a novel paradigm for morphogenetic robustness of the epithelial tissue by cell-turnover. The massive cell-turnover induced in the *M/+* wing pouch ensures normal wing growth during the abnormally prolonged larval period,

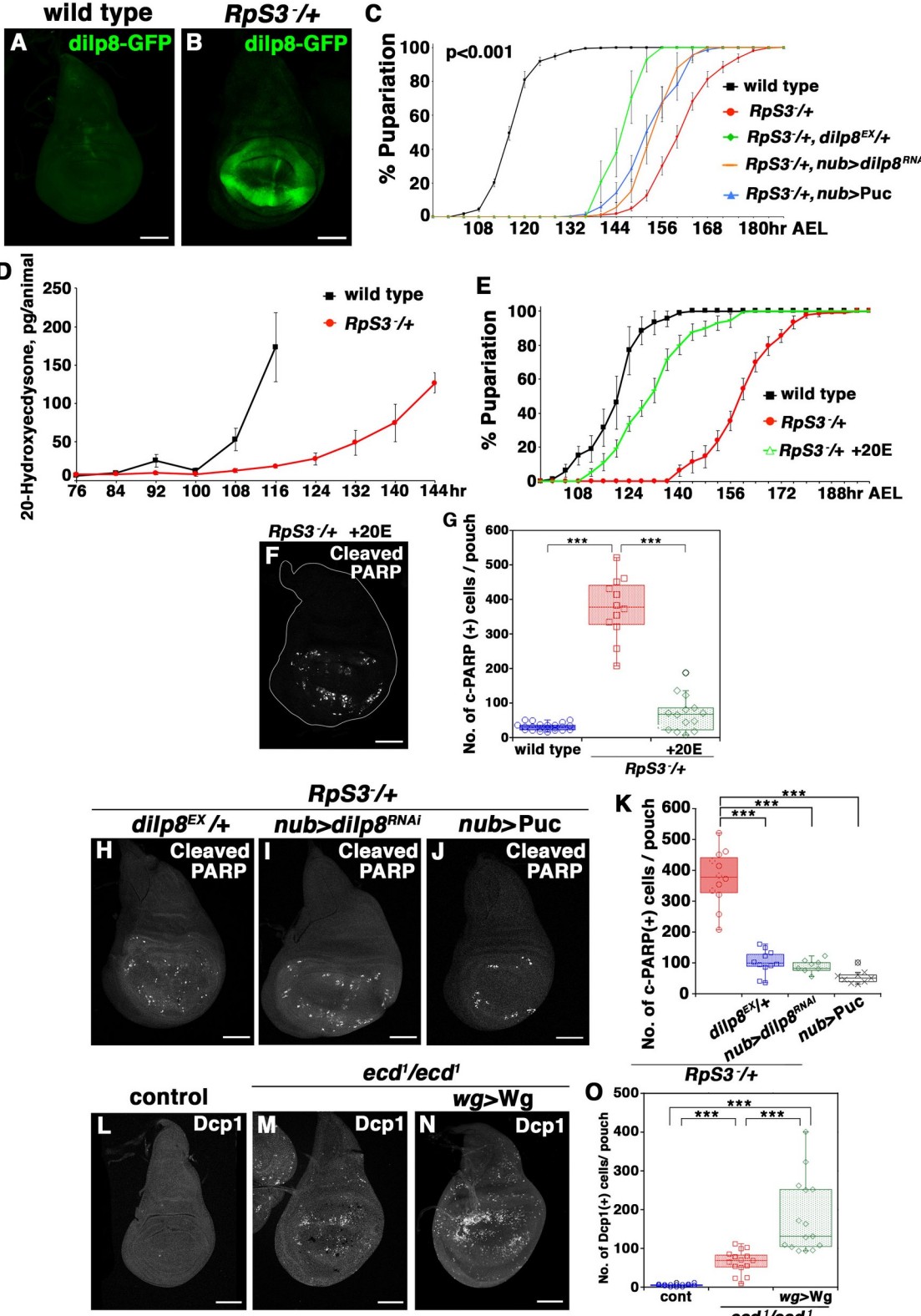

**Fig 4. JNK-Dilp8-mediated developmental delay is required for massive cell-turnover. (A and B)** The expression of dilp8 in wing discs of *dilp8::eGFP/+* (A) or *RpS3/+, Dilp8::eGFP/+* (B) flies were visualized with GFP (green). Scale bar, 100 μm. **(C)** Pupariation curves for wild-type (n = 390), *RpS3/+* (n = 666), *RpS3/+, dilp8^EX/+* (n = 46), *RpS3/+, nub>dilp8^RNAi* (n = 215), or *RpS3/+, nub>*Puc

(n = 302) flies. *p*>0.001, comparing *RpS3*/+ to the other flies (log-rank analysis). Error bars, SEM. **(D)** 20-Hydroxyecdysteroid titer was measured in wild-type or *RpS3*/+ larvae. **(E)** Pupariation curves of wild-type or *RpS3*/+ larvae that were fed food containing 0.75 mg/ml 20E or control. Error bars, SEM. **(F)** Dying cells were visualized by anti-cleaved PARP staining (white) in the wing discs of *RpS3*/+ larvae that were fed food containing 0.75 mg/ml 20E. Scale bar, 100 μm. **(G)** Boxplot with dots representing cleaved-PARP-positive dying cells per pouch in genotypes shown in (Fig 3A) (n = 22), (Fig 3B) (n = 12), and (F) (n = 14). Error bars, SEM; ***, p<0.001; non-parametric Mann-Whitney *U*-test. **(H-I)** Dying cells were detected by anti-cleaved PARP staining in the wing discs of *RpS3*/+, *dilp8^{EX}*/+ (H), *RpS3*/+, *nub-Gal4, UAS- dilp8-RNAi* (I), or *RpS3*/+, *nub-Gal4, UAS-Puc* (J) flies expressing CD8-PARP-Venus. Scale bar, 100 μm. **(K)** Boxplot with dots representing cleaved-PARP-positive dying cells per pouch in genotypes shown in (Fig 3B) (n = 12), (H) (n = 12), (I) (n = 9), and (J) (n = 9). Error bars, SEM; ***, p<0.001; non-parametric Mann-Whitney *U*-test. **(L-N)** Dying cells were visualized by anti-cleaved Dcp-1 staining in the wing discs (white). *ecd^1* is a temperature-sensitive *ecd* mutant allele that blocks biosynthesis of the active-form of the hormone 20-Hydroxyecdysone at 29˚C. Non-heat-shock treated flies was grown at 18˚C (L). For heat-shock treatment, *ecd^1*/*ecd^1* (M), or *ecd^1*/*ecd^1*, *wg-Gal4, UAS-Wg* (N) fly culture was transferred to 29˚C for 48 hours during the 3rd instar larval stage. **(O)** Boxplot with dots representing cleaved-Dcp-1-positive dying cells per pouch in genotypes shown in (L) (n = 14), (M) (n = 15) and (N) (n = 15). Error bars, SEM; ***, p<0.001; non-parametric Mann-Whitney *U*-test. Scale bar, 100 μm. Genotypes are as follows: *dilp8::eGFP*/+ (A), *RpS3^{Plac92}*/*dilp8::eGFP* (B), *nub-Gal4*/+; *UAS-CD8-PARP-Venus*/+ (black), *nub-Gal4*/+; *UAS-CD8-PARP-Venus, RpS3^{Plac92}*/+ (red), *nub-Gal4*/+; *UAS-CD8-PARP-Venus, RpS3^{Plac92}*/*dilp8^{EX}* (green), *nub-Gal4/UAS-dilp8-RNAi; UAS-CD8-PARP-Venus, RpS3^{Plac92}*/+ (orange), *nub-Gal4*/+; *UAS-CD8-PARP-Venus, RpS3^{Plac92}*/*UAS-Puc* (blue) (C), *w^{1118}* (D: wild-type), *FRT82B, Ubi-GFP, RpS3^{Plac92}*/+ (D; RpS3/+), *nub-Gal4*/+; *UAS-CD8-PARP-Venus*/+ (E: wild-type), *nub-Gal4*/+; *UAS-CD8-PARP-Venus, RpS3^{Plac92}*/+ (E: *RpS3*/+, F), *nub-Gal4*/+; *UAS-CD8-PARP-Venus, RpS3^{Plac92}*/*dilp8^{EX}* (H), *nub-Gal4*/+; *UAS-CD8-PARP-Venus, RpS3^{Plac92}*/*UAS-dilp8-RNAi* (I), *nub-Gal4*/+; *UAS-CD8-PARP-Venus, RpS3^{Plac92}*/*UAS-Puc* (J), *ecd^1*/*ecd^1* (L, M), and *wg-Gal4/UAS-Wg; ecd^1*/*ecd^1* (N).

possibly by creating a flexible cell death and proliferation platform to adjust cell numbers in the prospective wing blade, just like the process of tissue regeneration [53]. The fact that impairment of cell-turnover in *M*/+ animals induces phenotypic variations (Fig 2M and 2N)

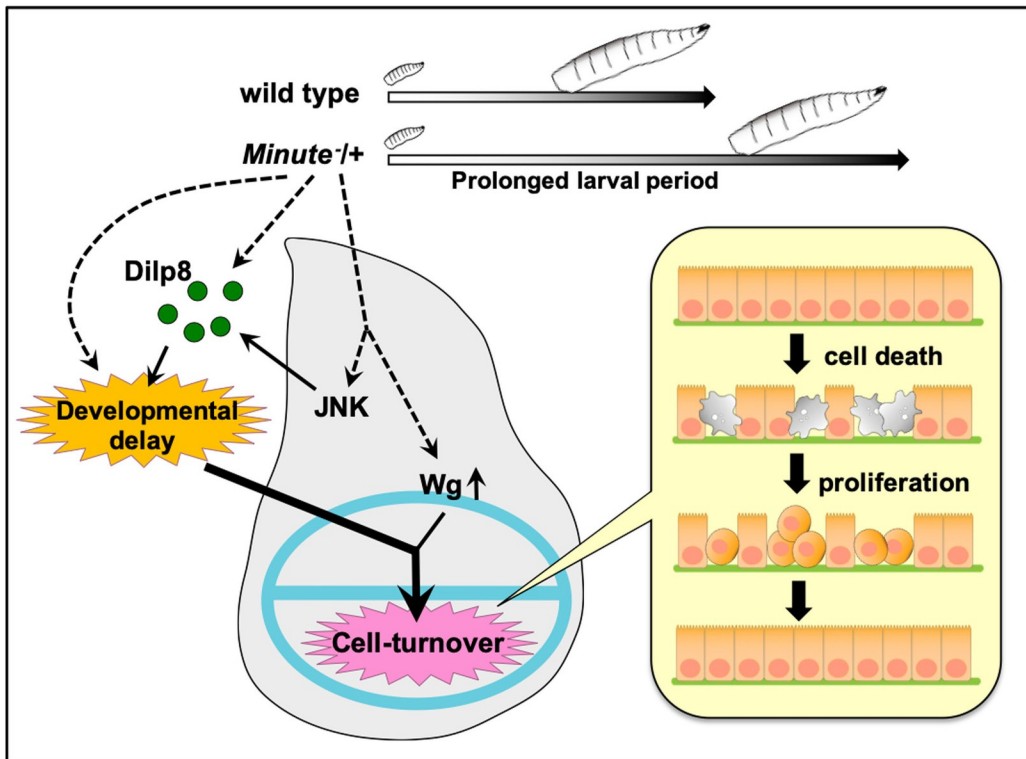

**Fig 5. A model for ensuring robust coordination of tissue growth in *M*/+ animals.** During prolonged larval period of *M*/+ animals, JNK signaling and Wg expression are upregulated in wing discs, which triggers a developmental delay and an aberrant Wg signaling gradient, respectively. These cooperate to induce massive cell turnover in the wing pouch, which permits normal wing morphogenesis.

suggests that cell-turnover may provide the flexibility to change developmental programs. Interestingly, it has been shown that a molecular chaperone heat-shock protein 90 (Hsp90) acts as a 'capacitor' for morphological evolution by buffering genetic variations during animal and plant development [54–56]. Thus, our findings in *M/+* flies suggest that massive cell-turnover may act as a buffering system for developmental time distortion, which could allow morphological variations. Such buffering system through cell-turnover could be also important for restore organ allometry, defined as the scaling of organ growth with body size [57], in response to developmental time distortion.

Our data suggest that massive cell-turnover in the *M/+* wing pouch is caused by an aberrant Wg signaling gradient generated by Wg upregulation. Wg upregulation gives rise to ectopic cell population with high Wg signaling activity in the wing pouch, likely triggering non-autonomous cell death of nearby cells with lower Wg signaling activity, which could be similar mechanism with cell competition. The prolonged larval period of *M/+* animals would further promote differences in Wg signaling activity between cells, as Wg signaling is continuously activated during larval development. Indeed, forced Wg upregulation using the endogenous *wg* promoter caused massive cell death when combined with developmental delay. While cell competition is thought to be involved in tissue homeostasis, tumor suppression, and the selection of fitter stem cells [58–60], its physiological role still remains elusive. Our data may suggest a role of cell competition in ensuring robust coordination of tissue growth during development.

The mechanism by which JNK activation in the *M/+* wing disc induces dilp8 expression is an important open question for future studies. It has been reported that JAK/STAT signaling acts downstream of JNK signaling to coordinate regenerative cell proliferation and developmental delay through upregulating *dilp8* expression during regeneration of the *Drosophila* fragmentated leg imaginal disc [61]. Interestingly, the activation pattern of JAK/STAT signaling in the *RpS3/+* wing disc is indistinguishable from that in the wild-type wing discs (S5A and S5B Fig), raising the possibility that another unknown signaling pathway or basal level of JAK/STAT signaling could regulate dilp8 expression, downstream of JNK in the *M/+* wing disc.

There are 88 *Drosophila* genes encoding cytoplasmic ribosomal proteins, and 65 of those heterozygous mutants (*M/+*) manifest phenotypes of a 2–3 day delay in larval development as well as short thin bristles. We found that different *M/+* strains showed similar massive cell death in their developing wing pouches, suggesting a general role for cell-turnover in morphogenetic robustness in ribosomal protein mutant animals. Intriguingly, heterozygous mutations in human genes encoding ribosomal proteins or ribosomal biogenesis factors are associated with genetic disorders with developmental defects, which are collectively called ribosomopathies [62]. For instance, heterozygous mutations in human *RpS19*, *RpS24*, *RpS17*, *RpL5*, *RpL11* or *RpL35A* are linked to diamond-Blackfan anaemia (DBA), and some DBA patients exhibit tissue-specific developmental defects, including limb defects, cleft palate, and abnormal heart development. Heterozygous mutations in the human *RpSA* gene have been identified in patients with isolated congenital asplenia (ICA), a disorder characterized by the absence of a spleen at birth [63]. Future studies on the molecular mechanism by which massive cell-turnover ensures morphogenetic robustness may help in understanding the etiology of ribosomopathies.

## Methods

### Fly strains

The following strains were used: *dilp8^ex* (Pierre Leopold), *10xSTAT-GFP* (Erika A. Bach), *UAS-CD8-PARP-Venus* (Yash Hiromi), *UAS-S/G2/M-Green* (Kaoru Sugimura), *hh-gal4*

(Tetsuya Tabata), *UAS-Dronc^{DN}* (Sharad Kumar), *wg-gal4* (Jean-Paul Vincent), *scrib^1* [64], *UAS-Puc* [65], *UAS-RpS3* [66], *RpS3^{Plac92}*, *RpL19^{K03704}*, *RpS17^4*, *dsh^3*, *nmo^{P1}* (*nmo-lacZ*), *nkd^{04869a}* (*nkd-lacZ*), *wg^{17en40cP1}* (*wg-lacZ*), *msn^{06946}* (*msn-lacZ*), *dilp8^{MI00727}* (eGFP trap), *UAS-p35*, *UAS-Wg*, *UAS-dsh-RNAi*, *Df(3L)H99*, *nub-gal4*, *en-gal4*, *wg^{Gla-1}*, *TRE-DsRed*, *act-gal4* (Bloomington Stock Center), *UAS-Arm^{S10}*, *ecd^1*, *wg^{l-8}*, *FRT82B*, *GMR-hid*, *CL3R* (the *Drosophila* Genomics and Genetic Resources, DGGR), *UAS-tcf-RNAi*, *UAS-dsh-RNAi* (National Institute of Genetics Stock Center), and *UAS-dilp8-RNAi* (Vienna *Drosophila* RNAi Center, VDRC).

## Histology

Larval tissues were stained with standard immunohistochemical procedures using rabbit anti-cleaved PARP antibody (Cell Signaling Technology, CST, 1:200), mouse anti-Wg antibody (Developmental Studies Hybridoma Bank (DSHB), 1:500), rabbit anti-phospho-histone H3 (Ser10) antibody (CST, 1:100), rabbit anti-β-galactosidase antibody (MP Biomedicals, 1:2000), mouse anti-β-galactosidase antibody (Sigma, 1:500), and rabbit anti-Cleaved *Drosophila* Dcp-1 (Asp216) antibody (CST, 1:100) for 3 days. For EdU labeling, larvae were dissected in Schneider's *Drosophila* medium (GIBCO) containing 5% FBS and incubated in the same media with 12.5 μM EdU for 30 min. Detection of EdU was performed with Click-iT Plus EdU Alexa Fluor 647 Imaging Kit (Thermo Fisher Scientific; C10640), according to the manufacture's instruction. Samples were mounted with DAPI-containing SlowFade Gold Antifade Reagent (Molecular Probes). Images were taken with a Leica SP5, SP8, or SPE confocal microscope.

## Quantification of EdU-positive area

For the analysis of EdU-positive cells, total area of EdU-positive cells and the size of the wing pouch area were automatically measured using Z-stacked images with Fiji and calculated with Microsoft Excel for Mac, as previously described [67].

## Detection of dying cells

To detect dying cells in the wing pouch, the activated-caspase-3 indicator CD8-PARP-Venus were expressed in the pouch under the control of *nub-gal4* driver. The anti-Dcp1 antibody was used in the experiments using the *wg-gal4* driver for overexpression of Wg to genetically enhance Wg gradient.

## Quantification of the relative number of phospho-Histone H3 positive cells or dying cells

Cleaved Dcp1 signals or phospho-histone H3 signals were taken as 3D projection images from the whole wing pouch. Cleaved PARP signals were acquired at the confocal plane with the highest signal. The number of signals in the wing pouch was automatically or manually measured with Fiji and calculated with Microsoft Excel for Mac.

## Time-lapse imaging and quantification

To monitor cell cycle, late 3rd instar larvae were dissected and cultured in Schneider's *Drosophila* medium (GIBCO) with 5% FBS and 2% low-melting agarose/PBS using the Glass Bottom Dish (MATSUNAMI). Imaging was conducted at 10-min intervals for 7 hours with a Leica SP8 confocal microscope [19,68]. For quantitative analysis, the number of dividing cells and duration time (min) between S/G2 and M phase were manually counted and measured in the apical surface of wing pouches (3–5 stacked images in 50 μm x 50 μm area). To observe

wing eversion, white-pupae were dissected and cultured in M3 Insect Medium (Sigma) with 2% FBS and 0.5% methyl-cellulose over 12 hours [68].

### Measurement of adult wings

Left and right wings of female flies were rinsed with xylene and mounted in a Canada Balsam (Nacalai Tesque). Images of the wings were captured using Leica M205C stereo microscope.

### Quantification of Wg and *wg-lacZ* signals

Wg or *wg-lacZ* signals were acquired with Leica SPE confocal microscope. Intensity was measured with Fiji and the intensity plot lines were generated using the data of average fluorescence intensities (n>6) in the area marked in pale green. Statistical analysis was done using Microsoft Excel.

### Developmental time analysis

For collecting embryos, flies were allowed to lay eggs for 2 hours at 25˚C on grape juice agar plates. 1st instar larvae were collected 24 hours after egg laying and were transferred to fresh yeast-based food. Developmental time from egg to pupariation was calculated in 4 hours intervals. Each experiment was performed more than three times. For adding 20E (20-Hydroxyecdysone; SIGMA), around twenty larvae were transferred to 20E-containing food (0.75 mg/ml) 92hr after egg laying.

### Ecdysteroid measurements

Ecdysteroid levels were measured using 20-Hydroxyecdysone Enzyme Immunoassay (EIA) kit (Bertin Bioreagent) according to the manufacturer's protocol. Briefly, ten larvae were frozen and homogenized with a Pellet Mixer (Treff Lab) in 250 ml methanol. Ecdysone was re-extracted twice with 125 ml methanol. 50 ml of the methanol extract was evaporated and dissolved in EIA Buffer. Further procedures were curried out according to the EIA protocol. The absorbance was read at 405 nm using ARVO X3 (PerkinElmer).

### Heat-shock treatment

*ecdysoneless* mutant flies were grown at 18˚C for 5 days and heat-shocked at 29˚C for 48 hours. Flies were dissected right after heat-shock treatment at wandering stage.

### Fly cultures with ecdysteroid synthesis defect

To inhibit the ecdysteroid synthesis, larvae were reared in the food with *erg-2* sterol mutant strain (a gift from Tomonori Katsuyama). Given that the *erg-2* gene encodes the Δ8-Δ7-sterol isomerase, flies feeding food with *erg-2* yeast impair production of a sufficient titer of ecdysteroid hormone necessary for pupation, leading to the developmental arrest in the larval stage. To collect embryos, flies were allowed to lay eggs for 8 hours at 25˚C on acetic acid agar plates. Embryos were rinsed, dechorinated with 3% bleach, and rewashed with MilliQ. Embryos were then put on a cover glass and transferred into *erg-2* Δ yeast food.

### Supporting information

**S1 Fig. Related to Fig 1. Massive cell-turnover occurs during wing development in *M*/+ animals. (A and B)** Wing disc of *RpS17*/+ (A) or *RpL19*/+ (B) flies were stained with anti-phospho-histone H3 (pH3) (Ser10) antibody (white). Wing pouches were marked by pale green.

Scale bar, 100 μm. **(C)** Boxplot with dots representing pH3 positive cells in the pouch in genotypes shown in (Fig 1A) (n = 34, number of wing pouches), (A) (n = 46) and (B) (n = 27). Error bars, SEM; ***, p<0.001; non-parametric Mann-Whitney *U*-test. **(D and E)** The activated-caspase-3 indicator CD8-PARP-Venus was expressed in *RpS17/+* (D) or *RpL19/+* (E) flies and dying cells in the wing disc were visualized by anti-cleaved PARP antibody. The nuclei were visualized by DAPI staining. Arrowheads indicate massive cell death occurred along the D/V axis in the wing pouch. Scale bar, 100 μm. **(F and G)** Developing wing disc of wild-type or *RpS3/+* flies are shown. Dying cells visualized by anti-cleaved Dcp-1 staining (white). Wg expression was visualized by anti-Wg staining (green). Flies were allowed to lay eggs for 2 hours at 25°C on yeast-based food. Larvae were dissected at the indicated time points after egg laying (AEL). All flies were third instar larvae. Scale bar, 100 μm. Genotypes are as follows: *nub-Gal4/+; UAS-CD8-PARP-Venus/RpS17$^4$* (A, D), *nub-Gal4/RpL19$^{K03704}$; UAS-CD8-PARP-Venus/+* (B, E). *SgsΔ3–GFP/+* (F), *SgsΔ3–GFP/FRT82B, Ubi-GFP, RpS3$^{Plac92}$* (G).
(TIF)

**S2 Fig. Related to Fig 2. Epithelial cell-turnover is essential for robust wing development in *M*/+ animals. (A-C)** Wing discs of wild-type (A), *RpS3/+* (B), or *RpS3/+, nub-Gal4*, UAS-p35 (C) flies. Nuclei were stained with DAPI. Scale bar, 100 μm. **(D and E)** Adult wings of *RpS17/+, nub-Gal4*, UAS-p35 (D), or *RpL19/+, nub-Gal4*, UAS-p35 (E) fly. Scale bar, 500 μm. **(F)** The rate of defective wings in the genotypes shown in (D) (n = 32) and (E) (n = 31). **(G and H)** Time-lapse of wing eversion in *RpS3/+* (G) or *RpS3/+, nub-Gal4*, UAS-p35 (H). Arrows indicate the direction of extension. Arrowheads indicate abnormal wrinkled structures. **(I)** The percentage of abnormal pupal wings in *RpS3/+* (n = 53) or *RpS3/+, nub-Gal4*, UAS-p35 (n = 36), which were classified according to the level of severity. Genotypes are as follows: *nub-Gal4/+; UAS-CD8-PARP-Venus/+* (A), *nub-Gal4/+; UAS-CD8-PARP-Venus, RpS3$^{Plac92}$/+* (B), *nub-Gal4/+; UAS-CD8-PARP-Venus, RpS3$^{Plac92}$/ UAS-p35* (C), *nub-Gal4/+; UAS-p35/RpS17$^4$* (D), *nub-Gal4/RpL19$^{K03704}$; UAS-p35/+* (E), *nub-Gal4/+; FRT82B, Ubi-GFP, RpS3$^{Plac92}$/+* (G), and *nub-Gal4/+; FRT82B, Ubi-GFP, RpS3$^{Plac92}$/UAS-p35* (H).
(TIF)

**S3 Fig. Related to Fig 3. Aberrant Wg signal gradient is essential for the induction of massive cell-turnover. (A-B')** RpS3 was overexpressed in the posterior compartment of wing discs of *RpS3/+, wg-lacZ/+* (A) flies using the *en-Gal4* driver. *wg* expression was visualized by anti-β-galactosidase staining (white). Arrowheads indicate relatively lower expression of *wg*. Scale bar, 100 μm. **(C)** Quantification of *wg* intensity in the pale green-marked area shown in (A) (n = 6) or (B) (n = 7). The intensity was automatically or manually measured using ImageJ (Fiji) software as described in the previous manuscript [67]. **(D-E')** RpS3 was overexpressed in the posterior compartment of wing discs of *RpS3/+* (D) flies using the *en-Gal4* driver. Wg expression was visualized by anti-Wg staining (white). Arrowheads indicate relatively lower expression of Wg. Scale bar, 100 μm. **(F)** Quantification of Wg intensity in the pale green-marked area shown in (D) (n = 8) or (E) (n = 7) are shown. The intensity was automatically or manually measured using ImageJ (Fiji) software as described in the previous manuscript [67]. **(G-J)** Dying cells were visualized by anti-cleaved PARP staining in *RpS3/+, dsh$^3$/+* (G), *RpS3/+, nub-Gal4, UAS-tcf-RNAi* (H), *RpS3/+, wg$^{Gla-1}$/+* (I), or *RpS3/+, nub-Gal4, UAS-Arm$^{S10}$, TG80$^{ts7}$* (J) wing disc. Massive cell death was significantly suppressed by reducing Wg signal gradient. Scale bar, 100 μm. **(K)** Boxplot with dots representing cleaved-PARP-positive dying cells per pouch in genotypes shown in (Fig 3B) (n = 12, number of wing pouches), (G) (n = 8), (H) (n = 15), (I) (n = 31), and (J) (n = 24). Error bars, SEM; ***, p<0.001; non-parametric Mann-Whitney U-test. Genotypes are as follows: *en-Gal4, UAS-GFP/wg$^{17en40cP1}$; RpS3$^{Plac92}$,*

*UAS-RpS3/+* (A), *en-Gal4, UAS-GFP/wg^{17en40cP1}; RpS3^{Plac92}/+* (B), *en-Gal4, UAS-GFP/+; RpS3^{Plac92}, UAS-RpS3/+* (D), *en-Gal4, UAS-GFP/+; RpS3^{Plac92}/+* (E), *dsh^3/+; nub-Gal4/+; UAS-CD8-PARP-Venus, RpS3^{Plac92}/+* (G), *nub-Gal4/UAS-tcf-RNAi; UAS-CD8-PARP-Venus, RpS3^{Plac92}/+* (H), *nub-Gal4/wg^{gla-1}; UAS-CD8-PARP-Venus, RpS3^{Plac92}/+* (I), and *UAS-Arm^{S10}/+; nub-Gal4/+; UAS-CD8-PARP-Venus, RpS3^{Plac92}/ TG80^{ts7}* (J).
(TIF)

**S4 Fig. Related to [Fig 4]. JNK-Dilp8-mediated developmental delay is required for massive cell-turnover. (A and B)** Wg was overexpressed using the endogenous *wg* promoter (*wg-Gal4*). Dying cells in the wing discs of wild-type (A) or *wg-Gal4*, UAS-Wg (B) flies were visualized by anti-cleaved Dcp-1 staining. Scale bar, 100 μm. **(C)** Boxplot with dots representing cleaved-Dcp-1-positive dying cells per wing pouch of genotypes shown in (A) (n = 17, number of wing pouches) and (B) (n = 16). Error bars, SEM; n.s, not significant; non-parametric Mann-Whiteney *U*-test. **(D)** The expression of dilp8 in wing discs of *RpS3/+, Dilp8::eGFP/+, nub-Gal4*, UAS-Puc flies were visualized with GFP (green). Scale bar, 100 μm. **(E-F')** RpS3 was overexpressed in the posterior compartment of the wing disc of *TRE-DsRed/+* (E) or *RpS3/+, TRE-DsRed/+* (F) flies using the *en-Gal4* driver. JNK-activated cells were visualized by *TRE-DsRed* reporter (white). **(G-H')** RpS3 was overexpressed in the posterior compartment of the wing discs of *msn-lacZ/+* flies (G) or *RpS3/+, msn-lacZ/+* (H) flies using the *en-Gal4* driver. JNK activity was visualized by anti-β-galactosidase staining (white). Scale bar, 100 μm. **(I and J)** Wing disc of *ecd^1/ecd^1* (I) or *ecd^1/ecd^1, wg-Gal4*, UAS-Wg (J) were stained with anti-phospho-histone H3 (pH3) (Ser10) antibody (white). Wing pouches were marked by pale green. *ecd^1* is a temperature-sensitive *ecd* mutant allele that blocks biosynthesis of the active-form of the hormone 20-Hydroxyecdysone at 29°C. For heat-shock treatment, fly culture was transferred to 29°C for 48 hours during the 3rd instar 1arval stage. Scale bar, 100 μm. **(K)** Boxplot with dots representing pH3 positive cells per pouch in genotypes shown in (I) (n = 13, number of wing pouches) and (J) (n = 13). Error bars, SEM; *, p<0.05; non-parametric Mann-Whitney *U*-test. **(L and M)** Wg was overexpressed using the endogenous *wg* promoter (*wg-Gal4*) in 3rd instar larvae fed with food containing *erg-2* mutant yeast. Dying cells in the wing discs of wild-type (L) or *wg-Gal4*, UAS-Wg (M) flies were visualized by anti-cleaved Dcp-1 staining. Scale bar, 100 μm. (**N**) Boxplot with dots representing dying cells of cleaved-Dcp-1-positive cells per pouch in genotypes shown in (A) (n = 17, number of wing pouches), (B) (n = 16), (L) (n = 17), and (M) (n = 14). Error bars, SEM; ***, p<0.001; **, p<0.01; non-parametric Mann-Whitney *U*-test. Scale bar, 100 μm. **(O and P)** Dying cells in the wing disc were visualized by anti-Dcp1 antibody staining (white). Wing disc of wild-type (O) or *wg-Gal4*, UAS-Wg (P) shown in flies bearing the eye disc in which *scrib* mutant clones were generated and surrounding wild-type cells were simultaneously removed by a combination of GMR-hid and a recessive cell-lethal mutation, CL3R. Scale bar, 100 μm. **(Q)** Boxplot with dots representing cleaved-Dcp-1-positive dying cells per pouch in genotypes shown in (O) (n = 14, number of wing pouches) and (P) (n = 7). Error bars, SEM; *, p<0.05; non-parametric Mann-Whitney *U*-test. Scale bar, 100 μm. Genotypes are as follows: *wg-Gal4/+* (A, L), *wg-Gal4/+;UAS-Wg/+* (B, M), *nub-Gal4/+; RpS3^{Plac92}, dilp8::eGFP/UAS-Puc* (D), *en-Gal4, UAS-GFP/+; TRE-DsRed/UAS-RpS3* (E), *en-Gal4, UAS-GFP/+; RpS3^{Plac92}, UAS-RpS3/ TRE-DsRed* (F), *en-Gal4, UAS-GFP/+; UAS-RpS3/ msn^{06946}* (G), *en-Gal4, UAS-GFP/+; RpS3^{Plac92}, UAS-RpS3/msn^{06946}* (H), *ecd^1/ecd^1* (I), *wg-Gal4/UAS-Wg; ecd^1/ ecd^1* (J), *eyFLP1/+;; FRT82B,scrib^1/FRT82B, GMR-hid.CL3R* (O), and *eyFLP1/+; wg-Gal4/UAS-Wg; FRT82B,scrib^1/FRT82B, GMR-hid.CL3R* (P).
(TIF)

**S5 Fig. The activation pattern of JAK/STAT signaling in the *M/+* wing disc is indistinguishable from that in the wild-type wing disc. (A-B')** The wild-type (A) or *RpS3/+* (B) wing

disc bearing the 10xSTAT-GFP reporter (green). The nuclei were visualized by DAPI staining (blue) Scale bar, 100 μm. Genotypes are as follows: *10xSTAT-GFP/+* (A) and *10xSTAT-GFP/ RpS3^{Plac92}* (B).
(TIF)

**S1 Movie. Live imaging of wing discs with the cell-cycle probe S/G2/M-Green.** The S/G2/ M-Green probe was expressed in the wing pouches of wild-type larvae. Wing discs were dissected and a part of the wing pouch was subjected to time-lapse imaging at 10-min intervals for 7 hours with a Leica SP8 confocal microscope. See *Methods* for details. Genotypes are as follows: *nub-Gal4/+; UAS-S/G2/M-Green/+*. Each experiment was replicated at least 3 times.
(MOV)

**S2 Movie. Live imaging of wing discs with the cell-cycle probe S/G2/M-Green.** The S/G2/ M-Green probe was expressed in the wing pouches of or *RpS3/+* larvae. Wing discs were dissected and a part of the wing pouch was subjected to time-lapse imaging at 10-min intervals for 7 hours with a Leica SP8 confocal microscope. See *Methods* for details. Genotypes are as follows: *nub-Gal4/+; UAS-S/G2/M-Green/RpS3^{Plac92}*. Each experiment was replicated at least 3 times.
(MOV)

**S3 Movie. Live imaging of wing disc eversion.** GFP-expressing wing discs of wild-type larvae were dissected and subjected to time-lapse imaging at 10-min intervals for over 12 hours with a Leica SP8 confocal microscope. See *Methods* for details. Genotypes are as follows: *FRT82B, Ubi-GFP/+*. Each experiment was replicated at least 3 times.
(MOV)

**S4 Movie. Live imaging of wing disc eversion.** GFP-expressing wing discs of *RpS3/+* larvae were dissected and subjected to time-lapse imaging at 10-min intervals for over 12 hours with a Leica SP8 confocal microscope. See *Methods* for details. Genotypes are as follows: *FRT82B, Ubi-GFP, RpS3^{Plac92}/+*. Each experiment was replicated at least 3 times.
(MOV)

**S5 Movie. Live imaging of wing disc eversion.** GFP-expressing wing discs of *RpS3/+*, *nub-Gal4*, UAS-p35 larvae were dissected and subjected to time-lapse imaging at 10-min intervals for over 12 hours with a Leica SP8 confocal microscope. See *Methods* for details. Genotypes are as follows: *nub-Gal4/+; FRT82B, Ubi-GFP, RpS3^{Plac92}/UAS-p35*. Each experiment was replicated at least 3 times.
(MOV)

## Acknowledgments

We thank Pierre Leopold, Laura Boulan, and John Vaughen for comments on the manuscript; Yayoi Wada, Aya Betsumiya, Kanako Nakagawa, Megumi Nakayama, Sayaka Kikuhara, Toshiharu Sawada, Kanako Baba, Miho Tanaka, Minoru Koijima for technical support; Carl Thummel, Vincent Henrich, Erika Bach, Yash Hiromi, Kaoru Sugimura, Tetsuya Tabata, Sharad Kumar, Jean-Paul Vincent, Masayuki Miura, Tian Xu, the Bloomington *Drosophila* Stock Center (Indiana), the Vienna *Drosophila* Resource Center (Vienna), the National Institute of Genetics Stock Center (Mishima), and the *Drosophila* Genomics and Genetic Resources (Kyoto) for fly stocks.

## Author Contributions

**Conceptualization:** Shizue Ohsawa, Tatsushi Igaki.

**Data curation:** Nanami Akai, Yukari Sando.

**Formal analysis:** Nanami Akai, Yukari Sando.

**Funding acquisition:** Shizue Ohsawa, Tatsushi Igaki.

**Investigation:** Nanami Akai, Shizue Ohsawa, Yukari Sando.

**Project administration:** Shizue Ohsawa, Tatsushi Igaki.

**Supervision:** Shizue Ohsawa.

**Validation:** Nanami Akai, Shizue Ohsawa, Yukari Sando.

**Visualization:** Nanami Akai, Shizue Ohsawa, Tatsushi Igaki.

**Writing – original draft:** Nanami Akai, Shizue Ohsawa, Tatsushi Igaki.

**Writing – review & editing:** Nanami Akai, Shizue Ohsawa, Yukari Sando, Tatsushi Igaki.

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
