## [Decision Letter · Decision Letter 0]

12 Jul 2020

Dear Tatsushi,

Thank you very much for submitting your Research Article entitled 'Epithelial cell-turnover ensures robust coordination of tissue growth in Drosophila ribosomal protein mutants' to PLOS Genetics. Your manuscript was fully evaluated at the editorial level and by independent peer reviewers. The reviewers appreciated the attention to an important problem, but raised some  concerns about the current manuscript. Based on the reviews, we will not be able to accept this version of the manuscript, but we would be willing to review again a revised version. We cannot, of course, promise publication at that time.

If you decide to revise the manuscript for further consideration at PLOS Genetics, please aim to resubmit within the next 60 days, unless it will take extra time to address the concerns of the reviewers, in which case we would appreciate an expected resubmission date by email to plosgenetics@plos.org.

[LINK]

We are sorry that we cannot be more positive about your manuscript at this stage. Please do not hesitate to contact us if you have any concerns or questions.

Yours sincerely,

Norbert Perrimon

Associate Editor

PLOS Genetics

Gregory P. Copenhaver

Editor-in-Chief

PLOS Genetics

Reviewer's Responses to Questions

**Comments to the Authors:**

Reviewer #1: This manuscript by Akai et al. reports a study of wing disc development in Minute Drosophila mutant larvae. The authors found abnormally high levels of both cell proliferation and cell death in the wings of these animals. They call this “massive cell turnover”. Dissection of this phenomenon showed that it is an instance of regeneration through apoptosis-induced compensatory proliferation, ensuring normal wing development through the known JNK-Wg axis. Furthermore, developmental delay through the also known JNK-Dilp8 axis contributes to the phenomenon. Overall, this is a well-written manuscript reporting a study that is both interesting and expertly conducted, with exquisite quantification of phenotypes. My major concerns have to do with the interpretation of the results, or sometimes lack of it, as I detail below.

1. It is not clear to me what is the ultimate reason that Minute animals show this increased cell death. The model the authors propose is that in Minute/+ animals cell death induces both compensatory proliferation through Wg and developmental delay through Dilp8, and that developmental delay somehow contributes to amplify the response. This is my take from the text. However, the authors are carefully and deliberately ambiguous about the relations among these elements in the model in Fig. 5, grouping proliferation and death together as “cell turnover”. “Minute” appears upstream in the diagram, but it is not clear whether this refers to Minute disc or Minute larva. Then, prolonged larval period is predicated of the larva, but it is not clear whether this is the same as the “developmental delay” downstream of Dilp8 or a different, more upstream event. I think a clear possibility is that the whole thing is a consequence of developmental delay and that developmental delay through JNK-Dilp8 is an amplifying event, whereas developmental delay of unknown origin (Dilp8 cannot account for all of it) is the ultimate cause of the whole thing. 4F shows a level of rescue of cell death by ecdysone in the Minute discs that is compatible with this. Furthermore, ecdysoneless mutants in 4M show an amount of cell death that is again compatible with this possibility. This should be explicitly acknowledged and discussed.

2. An alternative interpretation would be that Minute mutant discs have autonomously elevated cell death, intrinsically and independent of developmental delay. In that case, developmental delay induced by cell death would amplify cell death and cell proliferation, but would not be the upstream triggering event. The timing of cell death in Minute discs in Fig S1 suggests that autonomous causes are responsible for elevated cell death and developmental delay later contributes to amplifying it. My understanding is that the field assumes Minute mutant disc cells develop slower but fine. This I thought was the reason the authors repeatedly state that their results are “surprising”, but do not explain what is surprising about this. Do the authors agree? Is there anything in their data or in the literature suggesting that Minute cells die at a higher rate in a context-independent way? If that is the interpretation of the authors, there are important consequences for the field and the concept of cell competition itself that the authors should again discuss.

Other comments:

A. Line 260: “Our genetic study of M/+ mutants proposes a novel paradigm…” What happens in these discs seems to me very similar to the rn>egr model of regeneration that Iswar Hariharan’s lab established. This should be acknowledged and his papers cited.

B. Line 267: “Interestingly, it has been shown that a molecular chaperone heat-shock protein 90 (Hsp90) acts as a ‘capacitor’ for morphological evolution by buffering genetic variations during animal and plant development (52-54). Thus, our findings in M/+ flies suggest that massive cell-turnover may act as a buffering system for developmental time distortion, which could allow morphological variations.”

The results here can be understood in the conceptual framework of regeneration through apoptosis-induced proliferation plus injury-induced developmental delay. I really don’t understand the Hsp90 reference here and the relation with morphological evolution.

C. The authors in the abstract write “Minute/+ mutants”. They should write instead “Minute mutants”, later clarifying in the introduction that these are dominant haploinsufficient mutations and that the analysis corresponds to Minute heterozygous (or hemizygous) animals (M/+).

D. Expressions like “surprisingly”, “strikingly”, “interestingly” or “intriguingly” should be used more sparingly by the authors if they want them to achieve any effect.

E. Semi-quantitative words should be avoided:

Massive cell turnover, massive cell death

Line 89: much lower

Line 111: only occasionally

Line 244: slightly increased

F. Others:

Line 180: a phenomenon BY WHICH

Line 214: PARTIALLY suppressed

Line 250: number of DYING CELLS

Reviewer #2: The manuscript by Akai et al., provides a detailed molecular genetic analysis of the developmental delay phenotype caused by heterozygosity for ribosomal mutants in Drosophila. These mutations, collectively referred to as Minutes, are known to cause pronounced developmental delay, but with the exception of a thin bristle phenotype, produce largely normal adults. Akai et al., were curious how normally proportioned adults where produced by Minute heterozygous mutants even though they experience substantial developmental delay. In recent years it has been discovered that Minute mutants induce expression of the insulin/relaxin like factor dilp8. Dilp8 is a stress signal that attenuates the production of the steroid hormone ecdysone, the metamorphosis inducing factor, thus prolonging developmental time. Therefore, a simple explanation for the relatively normal adult body size and patterning of Minute mutants might have been that in Minute heterozygous larvae all mitotic cells grow and divide slowly as a result of a reduced ribosome pool. This leads induction of the dilp8 stress signal resulting in delayed metamorphosis. Thus, the slow growth rate is compensated by developmental delay leading to the formation of essential normal adults. What the authors found, however, was something much more intriguing. When they examined the proliferation rates of Minute heterozygous imaginal tissue, they discovered that it was actually significantly increase rather than decreased as the simple model might predict. Therefore, in order to achieve proper size and pattern, the enhanced proliferation must be compensated by some additional mechanism. The authors found that the compensatory mechanism was dramatic induction of apoptosis. They then go on to investigate the molecular mechanism responsible for the Ying and Yang of proliferation/apoptosis and found that found that Minute heterozygosity leads to up-regulation of Wingless at the D/V compartment which triggers cell-competition as a result of an altered Wg gradient. Intriguingly, they further found that simply altering the Wg gradient in wildtype background is not sufficient to induce the massive apoptosis but that it also requires the dip8 mediated developmental delay. When they artificially altered the Wg gradient and slowed developmental time in a wt background, then apoptosis was induced. The authors conclude that robust cell turnover helps maintaining tissue homeostasis by interfacing with developmental timing mechanisms.

Overall, I find this to be a remarkable set of observations which leads to many additional questions. Obviously, these could be the subject of future studies, but perhaps a few additional experiments could lend a bit of additional clarity to the results.

The authors investigate JNK signaling as a means of inducing dilp8 but recognize that there must be some additional cues. Several other inputs that regulate dilp8 expression have been identified including Hippo and Jak/Stat signaling, the transcription factor Xrp1 and the Ecdysone Receptor. Although the authors mention that Jak/Stat signaling does not seem to be involved, have the examined the requirement of these other factors and pathways?

Another key regulator of of imaginal disc growth is Dpp. Did the authors examine Dpp expression to see if it was altered in addition to Wg?

Another question concerns the degree to which the wing disc response is directly related to developmental time. The authors say it is perturbation of developmental time that is an important aspect of the process and to illustrate this they perturb timing by using the ecd ts mutant while over-expressing Wg in its endogenous pattern which produces strong apoptosis in the wing pouch. But, is it really timing or simply the basal level of ecdysone which is sensed by the disc? Can the authors see the same effect if they block ecdysone reception in the disc itself by either expressing dominant negative EcR or perhaps overexpress RNAi of ECI, the ecdysone importer. This would require using two different conditional expression systems but would add some clarity to the actual mechanism. Likewise, the ecd ts mutant, while convenient, is not the best way to specifically alter E production since it is a splicing factor that may affect many other processes in addition to ecdysone production. Again, the use of two different conditional systems, one to knockdown E production in the PG and the other to overexpress Wg would be cleaner than using the ecd ts mutation.

Another issue is whether heterozygosity for a Minute is only affecting cell turnover in the imaginal discs. Have the authors examined if other mitotic tissues such as the brain, or mitotic portions of the gut and/or the histoblasts also affected?

For the rescue experiments shown in Fig 4C, the authors find that heterozygosity for dilp8 partially overcomes the developmental delay produced by heterozygosity for RpS3 as does nub>dilp8 RNAi. However, the rescue observed when they feed ecdysone (Fig. 4E) is much better (less delay). Is this difference simply because dilp8 expression was not fully attenuated, or are there other signals involved? What happens to the delay if dilp8 RNAi is ubiquitously expressed?

Minor point: line 244 “…found that the number of cell death…” please reword

Reviewer #3: In this manuscript, the authors examine the Minute (M/+) phenotype in more detail using wing imaginal discs in Drosophila. M mutations affect ribosomal genes and cause a significant developmental delay. It was assumed that this delay is caused by reduced proliferation rates due to decreased ribosomal activity. However, contrary to this assumption, the authors found that at least one M mutation (RpS3-/+) is characterized by increased proliferation rates. Likewise, the authors observed significantly increased apoptosis in this and two other M/+ mutants. The authors then demonstrate that M/+ wing pouches exhibit greatly enhanced cell turnover that is induced by massive apoptosis and subsequent compensatory proliferation. In search for the apoptosis-inducing signal, the authors found that increased expression of Wg in an aberrant (steeper) gradient in the wing disc triggers apoptosis in M/+ discs. However, the steeper Wg gradient alone is not sufficient for apoptosis. The authors further found that the developmental delay of M/+ mutants contributes to the aberrant Wg gradient and apoptosis. Consistently, down-regulation of dilp8 which promotes the developmental delay, reduces the amount of apoptosis in M/+ discs. In addition, feeding ecdysone (E20) to M/+ larvae rescues the developmental delay and apoptosis. Finally, overexpression of wg under wg control which in wild-type larvae has very little effect on apoptosis, triggers strong apoptosis in ecdysoneless mutant larvae which are developmentally delayed independently of M/+.

This is a very interesting manuscript. It reports the very surprising discovery that M/+ wing discs are characterized by massive cell turnover due to increased apoptosis and compensatory proliferation. This cell turnover is dependent on the developmental delay and ensures developmental robustness in M/+ mutants. The data are very convincing and often different methods were applied to ensure consistency. The manuscript is very well written. I do have a few experimental suggestions and comments/questions to further improve the manuscript.

1. It would be good if the authors can show the increased proliferation in more than one M/+ mutant. They showed the increased cell death for 3 M/+ strains, but the proliferation phenotype was only shown for one.

2. If I understand this correctly, the aberrant steeper Wg gradient is the result of the continuous wg expression during the delay period in M/+ wing discs. If correct, it would follow that the massive apoptosis is only triggered during the period of the developmental delay (so quite late in larval development). However, the authors write that “cell death increased as development preceded and peaked during middle to late 3rd instar” (lines 117-118). Is that consistent with the dependence of apoptosis on the aberrant Wg gradient and the developmental delay? When do the authors detect the aberrant Wg gradient first in development?

3. Wg expression along the DV border is dependent on Notch signaling. Did the authors observe any abnormalities of Notch signaling in M/+ discs?

4. Is the enhanced cell turnover also observed in other discs such as eye discs? If so, would there be a similar dependence on Wg in this tissue?

5. In the ecd;wg>wg experiment where the authors detect increased apoptosis (Fig. 4N), is there also compensatory proliferation?

6. Explain in the manuscript, what the erg-2 mutant yeast in the fly food accomplishes.

**Have all data underlying the figures and results presented in the manuscript been provided?**

Reviewer #1: Yes

Reviewer #2: Yes

Reviewer #3: Yes

PLOS authors have the option to publish the peer review history of their article (what does this mean?). If published, this will include your full peer review and any attached files.

Reviewer #1: No

Reviewer #2: No

Reviewer #3: No

---

## [Decision Letter · Decision Letter 1]

7 Dec 2020

Dear Tatsushi,

We are pleased to inform you that your manuscript entitled "Epithelial cell-turnover ensures robust coordination of tissue growth in Drosophila ribosomal protein mutants" has been editorially accepted for publication in PLOS Genetics. Congratulations! Please note a few minor suggestions from one of the reviewer (see below).

Yours sincerely,

Norbert Perrimon

Associate Editor

PLOS Genetics

Gregory P. Copenhaver

Editor-in-Chief

PLOS Genetics

Comments from the reviewers (if applicable):

Reviewer's Responses to Questions

**Comments to the Authors:**

Reviewer #1: My concerns have been addressed, but, in the interest of clarity, I suggest rewording changed sentences as follows:

Line 108: It has been well documented by clonal analysis that the growth rate of M/+ cell clones is lower than of wild-type clones in developing wing discs (9). The lower growth rate of M/+ implied that M/+ cells simply have a lower cell division rate without any defects in proliferation and cell death.

SUGGESTION: It has been well documented by clonal analysis that the growth rate of M/+ cell clones is lower than THAT of wild-type clones in developing wing discs (9). The lower growth rate of M/+ ANIMALS HAS BEEN LARGELY ASSUMED TO STEM FROM a lower cell division rate OF M/+ cells, without any defects in proliferation and cell death.

Line 242: These data indicate that elevated JNK activity in the M/+ wing pouch causes Dilp8-mediated developmental delay in the M/+ larvae. The developmental delay in M/+ animals could be triggered by unknown origin in M/+ larvae and is amplified by the JNK-Dilp8 axis, since developmental delay was not fully rescued by blocking the JNK-Dilp8 axis in M/+ wing discs.

SUGGESTION: These data indicate that elevated JNK activity in the M/+ wing pouch causes Dilp8-mediated developmental delay in the M/+ larvae. The developmental delay in M/+ animals, HOWEVER, could be INITIALLY triggered by ADDITIONAL unknown CAUSES, since developmental delay was not fully rescued by blocking the JNK-Dilp8 axis in M/+ wing discs.

Reviewer #2: The authors addressed all my concerns

**Have all data underlying the figures and results presented in the manuscript been provided?**

Reviewer #1: Yes

Reviewer #2: Yes

PLOS authors have the option to publish the peer review history of their article (what does this mean?). If published, this will include your full peer review and any attached files.

Reviewer #1: No

Reviewer #2: **Yes: **Michael B O'Connor

**Data Deposition**

http://datadryad.org/submit?journalID=pgenetics&manu=PGENETICS-D-20-00918R1

**Press Queries**

---

## [Editor Report · Acceptance letter]

13 Jan 2021

PGENETICS-D-20-00918R1 

Epithelial cell-turnover ensures robust coordination of tissue growth in Drosophila ribosomal protein mutants 

Dear Dr Igaki, 

We are pleased to inform you that your manuscript entitled "Epithelial cell-turnover ensures robust coordination of tissue growth in Drosophila ribosomal protein mutants" has been formally accepted for publication in PLOS Genetics! Your manuscript is now with our production department and you will be notified of the publication date in due course.

With kind regards,

Melanie Wincott

PLOS Genetics

On behalf of:
